# A Probabilistic Formulation of Unsupervised Text Style Transfer

**Junxian He**[*], **Xinyi Wang**[*], **Graham Neubig**
Carnegie Mellon University
{junxianh,xinyiw1,gneubig}@cs.cmu.edu

**Taylor Berg-Kirkpatrick**
University of California San Diego
tberg@eng.ucsd.edu

## Abstract

We present a deep generative model for unsupervised text style transfer that unifies previously proposed non-generative techniques. Our probabilistic approach models non-parallel data from two domains as a partially observed parallel corpus. By hypothesizing a parallel latent sequence that generates each observed sequence, our model learns to transform sequences from one domain to another in a completely unsupervised fashion. In contrast with traditional generative sequence models (e.g. the HMM), our model makes few assumptions about the data it generates: it uses a recurrent language model as a prior and an encoder-decoder as a transduction distribution. While computation of marginal data likelihood is intractable in this model class, we show that amortized variational inference admits a practical surrogate. Further, by drawing connections between our variational objective and other recent unsupervised style transfer and machine translation techniques, we show how our probabilistic view can unify some known non-generative objectives such as backtranslation and adversarial loss. Finally, we demonstrate the effectiveness of our method on a wide range of unsupervised style transfer tasks, including sentiment transfer, formality transfer, word decipherment, author imitation, and related language translation. Across all style transfer tasks, our approach yields substantial gains over state-of-the-art non-generative baselines, including the state-of-the-art unsupervised machine translation techniques that our approach generalizes. Further, we conduct experiments on a standard unsupervised machine translation task and find that our unified approach matches the current state-of-the-art.[1]

## 1 Introduction

Text sequence transduction systems convert a given text sequence from one domain to another. These techniques can be applied to a wide range of natural language processing applications such as machine translation (Bahdanau et al., 2015), summarization (Rush et al., 2015), and dialogue response generation (Zhao et al., 2017). In many cases, however, parallel corpora for the task at hand are scarce. Therefore, *unsupervised* sequence transduction methods that require only non-parallel data are appealing and have been receiving growing attention (Bannard & Callison-Burch, 2005; Ravi & Knight, 2011; Mizukami et al., 2015; Shen et al., 2017; Lample et al., 2018; 2019). This trend is most pronounced in the space of text *style transfer* tasks where parallel data is particularly challenging to obtain (Hu et al., 2017; Shen et al., 2017; Yang et al., 2018). Style transfer has historically referred to sequence transduction problems that modify superficial properties of text – i.e. style rather than content.[2] We focus on a standard suite of style transfer tasks, including formality transfer (Rao & Tetreault, 2018), author imitation (Xu et al., 2012), word decipherment (Shen et al., 2017), sentiment transfer (Shen et al., 2017), and related language translation (Pourdamghani & Knight, 2017). General unsupervised translation has not typically been considered style transfer, but for the purpose of comparison we also conduct evaluation on this task (Lample et al., 2017).

---

[*]Equal Contribution.

[1]Code and data are available at https://github.com/cindyxinyiwang/deep-latent-sequence-model.

[2]Notably, some tasks we evaluate on do change content to some degree, such as sentiment transfer, but for conciseness we use the term "style transfer" nonetheless.

Recent work on unsupervised text style transfer mostly employs non-generative or non-probabilistic modeling approaches. For example, Shen et al. (2017) and Yang et al. (2018) design adversarial discriminators to shape their unsupervised objective – an approach that can be effective, but often introduces training instability. Other work focuses on directly designing unsupervised training objectives by incorporating intuitive loss terms (e.g. backtranslation loss), and demonstrates state-of-the-art performance on unsupervised machine translation (Lample et al., 2018; Artetxe et al., 2019) and style transfer (Lample et al., 2019). However, the space of possible unsupervised objectives is extremely large and the underlying modeling assumptions defined by each objective can only be reasoned about indirectly. As a result, the process of designing such systems is often heuristic.

In contrast, probabilistic models (e.g. the noisy channel model (Shannon, 1948)) define assumptions about data more explicitly and allow us to reason about these assumptions during system design. Further, the corresponding objectives are determined naturally by principles of probabilistic inference, reducing the need for empirical search directly in the space of possible objectives. That said, classical probabilistic models for unsupervised sequence transduction (e.g. the HMM or semi-HMM) typically enforce overly strong independence assumptions about data to make exact inference tractable (Knight et al., 2006; Ravi & Knight, 2011; Pourdamghani & Knight, 2017). This has restricted their development and caused their performance to lag behind unsupervised neural objectives on complex tasks. Luckily, in recent years, powerful variational approximation techniques have made it more practical to train probabilistic models without strong independence assumptions (Miao & Blunsom, 2016; Yin et al., 2018). Inspired by this, we take a new approach to unsupervised style transfer.

We directly define a generative probabilistic model that treats a non-parallel corpus in two domains as a partially observed parallel corpus. Our model makes few independence assumptions and its true posterior is intractable. However, we show that by using amortized variational inference (Kingma & Welling, 2013), a principled probabilistic technique, a natural unsupervised objective falls out of our modeling approach that has many connections with past work, yet is different from all past work in specific ways. In experiments across a suite of unsupervised text style transfer tasks, we find that the natural objective of our model actually outperforms all manually defined unsupervised objectives from past work, supporting the notion that probabilistic principles can be a useful guide even in deep neural systems. Further, in the case of unsupervised machine translation, our model matches the current state-of-the-art non-generative approach.

## 2 UNSUPERVISED TEXT STYLE TRANSFER

We first overview text style transfer, which aims to transfer a text (typically a single sentence or a short paragraph – for simplicity we refer to simply "sentences" below) from one domain to another while preserving underlying content. For example, formality transfer (Rao & Tetreault, 2018) is the task of transforming the tone of text from informal to formal without changing its content. Other examples include sentiment transfer (Shen et al., 2017), word decipherment (Knight et al., 2006), and author imitation (Xu et al., 2012). If parallel examples were available from each domain (i.e. the training data is a bitext consisting of pairs of sentences from each domain), supervised techniques could be used to perform style transfer (e.g. attentional Seq2Seq (Bahdanau et al., 2015) and Transformer (Vaswani et al., 2017)). However, for most style transfer problems, only non-parallel corpora (one corpus from each domain) can be easily collected. Thus, work on style transfer typically focuses on the more difficult unsupervised setting where systems must learn from non-parallel data alone.

The model we propose treats an observed non-parallel text corpus as a partially observed parallel corpus. Thus, we introduce notation for both observed text inputs and those that we will treat as latent variables. Specifically, we let $X = \{x^{(1)}, x^{(2)}, \cdots, x^{(m)}\}$ represent observed data from domain $\mathcal{D}_1$, while we let $Y = \{y^{(m+1)}, y^{(m+2)}, \cdots, y^{(n)}\}$ represent observed data from domain $\mathcal{D}_2$. Corresponding indices represent parallel sentences. Thus, none of the observed sentences share indices. In our model, we introduce latent sentences to complete the parallel corpus. Specifically, $\bar{X} = \{\bar{x}^{(m+1)}, \bar{x}^{(m+2)}, \cdots, \bar{x}^{(n)}\}$ represents the set of latent parallel sentences in $\mathcal{D}_1$, while $\bar{Y} = \{\bar{y}^{(1)}, \bar{y}^{(2)}, \cdots, \bar{y}^{(m)}\}$ represents the set of latent parallel sentences in $\mathcal{D}_2$. Then the goal of unsupervised text transduction is to infer these latent variables conditioned the observed non-parallel corpora; that is, to learn $p(\bar{y}|x)$ and $p(\bar{x}|y)$.

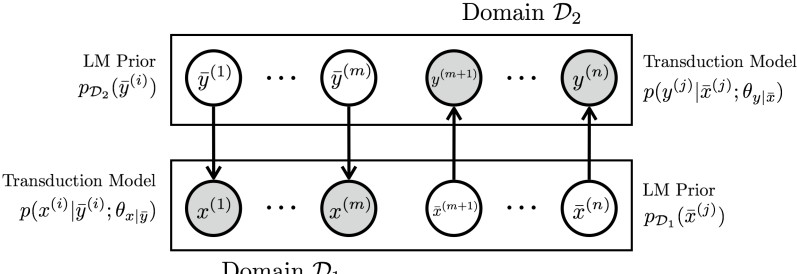

Figure 1: Proposed graphical model for style transfer via bitext completion. Shaded circles denote the observed variables and unshaded circles denote the latents. The generator is parameterized as an encoder-decoder architecture and the prior on the latent variable is a pretrained language model.

## 3 THE DEEP LATENT SEQUENCE MODEL

First we present our generative model of bitext, which we refer to as a deep latent sequence model. We then describe unsupervised learning and inference techniques for this model class.

### 3.1 MODEL STRUCTURE

Directly modeling $p(\bar{y}|x)$ and $p(\bar{x}|y)$ in the unsupervised setting is difficult because we never directly observe parallel data. Instead, we propose a generative model of the complete data that defines a joint likelihood, $p(X, \bar{X}, Y, \bar{Y})$. In order to perform text transduction, the unobserved halves can be treated as latent variables: they will be marginalized out during learning and inferred via posterior inference at test time.

Our model assumes that each observed sentence is generated from an unobserved parallel sentence in the opposite domain, as depicted in Figure 1. Specifically, each sentence $x^{(i)}$ in domain $\mathcal{D}_1$ is generated as follows: First, a latent sentence $\bar{y}^{(i)}$ in domain $\mathcal{D}_2$ is sampled from a prior, $p_{\mathcal{D}_2}(\bar{y}^{(i)})$. Then, $x^{(i)}$ is sampled conditioned on $\bar{y}^{(i)}$ from a transduction model, $p(x^{(i)}|\bar{y}^{(i)})$. Similarly, each observed sentence $y^{(j)}$ in domain $\mathcal{D}_2$ is generated conditioned on a latent sentence, $\bar{x}^{(j)}$, in domain $\mathcal{D}_1$ via the opposite transduction model, $p(y^{(j)}|\bar{x}^{(j)})$, and prior, $p_{\mathcal{D}_1}(\bar{x}^{(j)})$. We let $\theta_{x|\bar{y}}$ and $\theta_{y|\bar{x}}$ represent the parameters of the two transduction distributions respectively. We assume the prior distributions are pretrained on the observed data in their respective domains and therefore omit their parameters for simplicity of notation. Together, this gives the following joint likelihood:

$$p(X, \bar{X}, Y, \bar{Y}; \theta_{x|\bar{y}}, \theta_{y|\bar{x}}) = \left(\prod_{i=1}^{m} p\big(x^{(i)}|\bar{y}^{(i)}; \theta_{x|\bar{y}}\big) p_{\mathcal{D}_2}\big(\bar{y}^{(i)}\big)\right) \left(\prod_{j=m+1}^{n} p\big(y^{(j)}|\bar{x}^{(j)}; \theta_{y|\bar{x}}\big) p_{\mathcal{D}_1}\big(\bar{x}^{(j)}\big)\right)$$

(1)

The log marginal likelihood of the data, which we will approximate during training, is:

$$\log p(X, Y; \theta_{x|\bar{y}}, \theta_{y|\bar{x}}) = \log \sum_{\bar{X}} \sum_{\bar{Y}} p(X, \bar{X}, Y, \bar{Y}; \theta_{x|\bar{y}}, \theta_{y|\bar{x}})$$

(2)

Note that if the two transduction models share no parameters, the training problems for each observed domain are independent. Critically, we introduce parameter sharing through our variational inference procedure, which we describe in more detail in Section 3.2.

**Architecture:** Since we would like to be able to model a variety of transfer tasks, we choose a parameterization for our transduction distributions that makes no independence assumptions. Specifically, we employ an encoder-decoder architecture based on the standard attentional Seq2Seq model which has been shown to be successful across various tasks (Bahdanau et al., 2015; Rush et al., 2015). Similarly, our prior distributions for each domain are parameterized as recurrent language models which, again, make no independence assumptions. In contrast, traditional unsupervised generative sequence models typically make strong independence assumptions to enable exact inference (e.g. the HMM makes a Markov assumption on the latent sequence and emissions are one-to-one). Our model is more flexible, but exact inference via dynamic programming will be intractable. We address this problem in the next section.

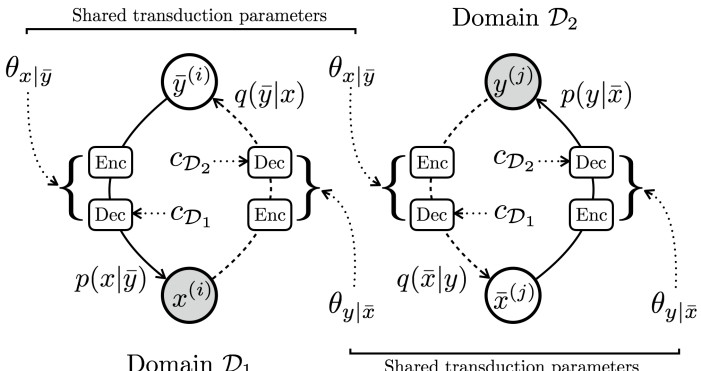

Figure 2: Depiction of amortized variational approximation. Distributions $q(\bar{y}|x)$ and $q(\bar{x}|y)$ represent inference networks that approximate the model's true posterior. Critically, parameters are shared between the generative model and inference networks to tie the learning problems for both domains.

## 3.2 LEARNING

Ideally, learning should directly optimize the log data likelihood, which is the marginal of our model shown in Eq. 2. However, due to our model's neural parameterization which does not factorize, computing the data likelihood cannot be accomplished using dynamic programming as can be done with simpler models like the HMM. To overcome the intractability of computing the true data likelihood, we adopt amortized variational inference (Kingma & Welling, 2013) in order to derive a surrogate objective for learning, the evidence lower bound (ELBO) on log marginal likelihood[3] :

$$
\begin{aligned}
&\log p(X, Y; \theta_{x|\bar{y}}, \theta_{y|\bar{x}}) \\
&\geq \mathcal{L}_{\text{ELBO}}(X, Y; \theta_{x|\bar{y}}, \theta_{y|\bar{x}}, \phi_{\bar{x}|y}, \phi_{\bar{y}|x}) \\
&= \sum_i \left[ \mathbb{E}_{q(\bar{y}|x^{(i)}; \phi_{\bar{y}|x})}[\log p(x^{(i)}|\bar{y}; \theta_{x|\bar{y}})] - D_{\text{KL}}\big(q(\bar{y}|x^{(i)}; \phi_{\bar{y}|x})||p_{\mathcal{D}_2}(\bar{y})\big) \right] \\
&+ \sum_j \underbrace{\left[ \mathbb{E}_{q(\bar{x}|y^{(j)}; \phi_{\bar{x}|y})}[\log p(y^{(j)}|\bar{x}; \theta_{y|\bar{x}})] \right.}_{\text{Reconstruction likelihood}} - \underbrace{\left. D_{\text{KL}}\big(q(\bar{x}|y^{(j)}; \phi_{\bar{x}|y})||p_{\mathcal{D}_1}(\bar{x})\big) \right]}_{\text{KL regularizer}}
\end{aligned}
\tag{3}
$$

The surrogate objective introduces $q(\bar{y}|x^{(i)}; \phi_{\bar{y}|x})$ and $q(\bar{x}|y^{(j)}; \phi_{\bar{x}|y})$, which represent two separate inference network distributions that approximate the model's true posteriors, $p(\bar{y}|x^{(i)}; \theta_{x|\bar{y}})$ and $p(\bar{x}|y^{(j)}; \theta_{y|\bar{x}})$, respectively. Learning operates by jointly optimizing the lower bound over both variational and model parameters. Once trained, the variational posterior distributions can be used directly for style transfer. The KL terms in Eq. 3, that appear naturally in the ELBO objective, can be intuitively viewed as regularizers that use the language model priors to bias the induced sentences towards the desired domains. Amortized variational techniques have been most commonly applied to continuous latent variables, as in the case of the variational autoencoder (VAE) (Kingma & Welling, 2013). Here, we use this approach for inference over discrete sequences, which has been shown to be effective in related work on a semi-supervised task (Miao & Blunsom, 2016).

**Inference Network and Parameter Sharing:** Note that the approximate posterior on one domain aims to learn the *reverse* style transfer distribution, which is exactly the goal of the generative distribution in the opposite domain. For example, the inference network $q(\bar{y}|x^{(i)}; \phi_{\bar{y}|x})$ and the generative distribution $p(y|\bar{x}^{(i)}; \theta_{y|\bar{x}})$ both aim to transform $\mathcal{D}_1$ to $\mathcal{D}_2$. Therefore, we use the same architecture for each inference network as used in the transduction models, and tie their parameters: $\phi_{\bar{x}|y} = \theta_{x|\bar{y}}, \phi_{\bar{y}|x} = \theta_{y|\bar{x}}$. This means we learn only two encoder-decoders overall – which are parameterized by $\theta_{x|\bar{y}}$ and $\theta_{y|\bar{x}}$ respectively – to represent two directions of transfer. In addition to reducing the number of learnable parameters, this parameter tying couples the learning problems for both domains and allows us to jointly learn from the full data. Moreover, inspired by recent work that

---

[3]Note that in practice, we add a weight $\lambda$ (the same to both domains) to the KL term in ELBO since the regularization strength from the pretrained language model varies depending on the datasets, training data size, or language model structures. Such reweighting has proven necessary in previous work that is trained with ELBO (Bowman et al., 2016; Miao & Blunsom, 2016; Yin et al., 2018).

builds a universal Seq2Seq model to translate between different language pairs (Johnson et al., 2017), we introduce further parameter tying between the two directions of transduction: the same encoder is employed for both $x$ and $y$, and a domain embedding $c$ is provided to the same decoder to specify the transfer direction, as shown in Figure 2. Ablation analysis in Section 5.3 suggests that parameter sharing is important to achieve good performance.

**Approximating Gradients of ELBO:** The reconstruction and KL terms in Eq. 3 still involve intractable expectations due to the marginalization over the latent sequence, thus we need to approximate their gradients. Gumbel-softmax (Jang et al., 2017) and REINFORCE (Sutton et al., 2000) are often used as stochastic gradient estimators in the discrete case. Since the latent text variables have an extremely large domain, we find that REINFORCE-based gradient estimates result in high variance. Thus, we use the Gumbel-softmax straight-through estimator to backpropagate gradients from the KL terms.[4] However, we find that approximating gradients of the reconstruction loss is much more challenging – both the Gumbel-softmax estimator and REINFORCE are unable to outperform a simple *stop-gradient* method that does not back-propagate the gradient of the latent sequence to the inference network. This confirms a similar observation in previous work on unsupervised machine translation (Lample et al., 2018). Therefore, we use greedy decoding without recording gradients to approximate the reconstruction term.[5] Note that the inference networks still receive gradients from the prior through the KL term, and their parameters are shared with the decoders which do receive gradients from reconstruction. We consider this to be the best empirical compromise at the moment.

**Initialization.** Good initialization is often necessary for successful optimization of unsupervised learning objectives. In preliminary experiments, we find that the encoder-decoder structure has difficulty generating realistic sentences during the initial stages of training, which usually results in a disastrous local optimum. This is mainly because the encoder-decoder is initialized randomly and there is no direct training signal to specify the desired latent sequence in the unsupervised setting. Therefore, we apply a self-reconstruction loss $\mathcal{L}_{\text{rec}}$ at the initial epochs of training. We denote the output the encoder as $e(\cdot)$ and the decoder distribution as $p_{\text{dec}}$, then

$$\mathcal{L}_{\text{rec}} = -\alpha \cdot \sum_i [p_{\text{dec}}(e(x^{(i)}), c_x)] - \alpha \cdot \sum_j [p_{\text{dec}}(e(y^{(j)}), c_y)], \tag{4}$$

$\alpha$ decays from 1.0 to 0.0 linearly in the first $k$ epochs. $k$ is a tunable parameter and usually less than 3 in all our experiments.

## 4 CONNECTION TO RELATED WORK

Our probabilistic formulation can be connected with recent advances in unsupervised text transduction methods. For example, back translation loss (Sennrich et al., 2016) plays an important role in recent unsupervised machine translation (Artetxe et al., 2018; Lample et al., 2018; Artetxe et al., 2019) and unsupervised style transfer systems (Lample et al., 2019). In order to incorporate back translation loss the source language $x$ is translated to the target language $y$ to form a pseudo-parallel corpus, then a translation model from $y$ to $x$ can be learned on this pseudo bitext just as in supervised setting. While back translation was often explained as a data augmentation technique, in our probabilistic formulation it appears naturally with the ELBO objective as the reconstruction loss term.

Some previous work has incorporated a pretrained language models into neural semi-supervised or unsupervised objectives. He et al. (2016) uses the log likelihood of a pretrained language model as the reward to update a supervised machine translation system with policy gradient. Artetxe et al. (2019) utilize a similar idea for unsupervised machine translation. Yang et al. (2018) employed a similar approach, but interpret the LM as an adversary, training the generator to fool the LM. We show how our ELBO objective is connected with these more heuristic LM regularizers by expanding the KL loss term (assume $x$ is observed):

$$D_{\text{KL}}(q(\bar{y}|x) || p_{\mathcal{D}_2}(\bar{y})) = -H_q - \mathbb{E}_q[\log p_{\mathcal{D}_2}(\bar{y})], \tag{5}$$

Note that the loss used in previous work does not include the negative entropy term, $-H_q$. Our objective results in this additional "regularizer", the negative entropy of the transduction distribution, $-H_q$. Intuitively, $-H_q$ helps avoid a peaked transduction distribution, preventing the transduction

---

[4] We use one sample to approximate the expectations.

[5] We compare greedy and sampling decoding in Section 5.3.

from constantly generating similar sentences to satisfy the language model. In experiments we will show that this additional regularization is important and helps bypass bad local optima and improve performance. These important differences with past work suggest that a probabilistic view of the unsupervised sequence transduction may provide helpful guidance in determining effective training objectives.

# 5 EXPERIMENTS

We test our model on five style transfer tasks: sentiment transfer, word substitution decipherment, formality transfer, author imitation, and related language translation. For completeness, we also evaluate on the task of general unsupervised machine translation using standard benchmarks.

We compare with the unsupervised machine translation model (UNMT) which recently demonstrated state-of-the-art performance on transfer tasks such as sentiment and gender transfer (Lample et al., 2019).[6] To validate the effect of the negative entropy term in the KL loss term Eq. 5, we remove it and train the model with a back-translation loss plus a language model negative log likelihood loss (which we denote as BT+NLL) as an ablation baseline. For each task, we also include strong baseline numbers from related work if available. For our method we select the model with the best validation ELBO, and for UNMT or BT+NLL we select the model with the best back-translation loss. Complete model configurations and hyperparameters can be found in Appendix A.1.

## 5.1 DATASETS AND EXPERIMENT SETUP

**Word Substitution Decipherment.** Word decipherment aims to uncover the plain text behind a corpus that was enciphered via word substitution where word in the vocabulary is mapped to a unique type in a cipher dictionary (Dou & Knight, 2012; Shen et al., 2017; Yang et al., 2018). In our formulation, the model is presented with a non-parallel corpus of English plaintext and the ciphertext. We use the data in (Yang et al., 2018) which provides 200K sentences from each domain. While previous work (Shen et al., 2017; Yang et al., 2018) controls the difficulty of this task by varying the percentage of words that are ciphered, we directly evaluate on the most difficult version of this task – 100% of the words are enciphered (i.e. no vocabulary sharing in the two domains). We select the model with the best unsupervised reconstruction loss, and evaluate with BLEU score on the test set which contains 100K parallel sentences. Results are shown in Table 2.

**Sentiment Transfer.** Sentiment transfer is a task of paraphrasing a sentence with a different sentiment while preserving the original content. Evaluation of sentiment transfer is difficult and is still an open research problem (Mir et al., 2019). Evaluation focuses on three aspects: attribute control, content preservation, and fluency. A successful system needs to perform well with respect to all three aspects. We follow prior work by using three automatic metrics (Yang et al., 2018; Lample et al., 2019): classification accuracy, self-BLEU (BLEU of the output with the original sentence as the reference), and the perplexity (PPL) of each system's output under an external language model. We pretrain a convolutional classifier (Kim, 2014) to assess classification accuracy, and use an LSTM language model pretrained on each domain to compute the PPL of system outputs.

We use the Yelp reviews dataset collected by Shen et al. (2017) which contains 250K negative sentences and 380K positive sentences. We also use a small test set that has 1000 human-annotated parallel sentences introduced in Li et al. (2018). We denote the positive sentiment as domain $\mathcal{D}_1$ and the negative sentiment as domain $\mathcal{D}_2$. We use Self-BLEU and BLEU to represent the BLEU score of the output against the original sentence and the reference respectively. Results are shown in Table 1.

**Formality Transfer.** Next, we consider a harder task of modifying the formality of a sequence. We use the GYAFC dataset (Rao & Tetreault, 2018), which contains formal and informal sentences from two different domains. In this paper, we use the Entertainment and Music domain, which has about 52K training sentences, 5K development sentences, and 2.5K test sentences. This dataset actually contains parallel data between formal and informal sentences, which we use only for evaluation. We follow the evaluation of sentiment transfer task and test models on three axes. Since the test set is

---

[6]The model they used is slightly different from the original model of Lample et al. (2018) in certain details – e.g. the addition of a pooling layer after attention. We re-implement their model in our codebase for fair comparison and verify that our re-implementation achieves performance competitive with the original paper.

Table 1: Results on the sentiment transfer, author imitation, and formality transfer. We list the PPL of pretrained LMs on the test sets of both domains. We only report Self-BLEU on the sentiment task to compare with existing work.

| Task | Model | Acc. | BLEU | Self-BLEU | $PPL_{\mathcal{D}_1}$ | $PPL_{\mathcal{D}_2}$ |
|------|-------|------|------|-----------|-----------|-----------|
| **Sentiment** | Test Set | - | - | - | 31.97 | 21.87 |
| | Shen et al. (2017) | 79.50 | 6.80 | 12.40 | 50.40 | 52.70 |
| | Hu et al. (2017) | 87.70 | - | **65.60** | 115.60 | 239.80 |
| | Yang et al. (2018) | 83.30 | 13.40 | 38.60 | 30.30 | 42.10 |
| | UNMT | 87.17 | 16.99 | 44.88 | 26.53 | 35.72 |
| | BT+NLL | **88.36** | 12.36 | 31.48 | 8.75 | 12.82 |
| | Ours | 87.90 | **18.67** | 48.38 | 27.75 | 35.61 |
| **Author Imitation** | Test Set | - | - | - | 132.95 | 85.25 |
| | UNMT | 80.23 | 7.13 | - | 40.11 | 39.38 |
| | BT+NLL | 76.98 | 10.80 | - | 61.70 | 65.51 |
| | Ours | **81.43** | **10.81** | - | 49.62 | 44.86 |
| **Formality** | Test Set | - | - | - | 71.30 | 135.50 |
| | UNMT | 78.06 | 16.11 | - | 26.70 | 10.38 |
| | BT+NLL | **82.43** | 8.57 | - | 6.57 | 8.21 |
| | Ours | 80.46 | **18.54** | - | 22.65 | 17.23 |

a parallel corpus, we only compute reference BLEU and ignore self-BLEU. We use $\mathcal{D}_1$ to denote formal text, and $\mathcal{D}_2$ to denote informal text. Results are shown in Table 1.

**Author Imitation.** Author imitation is the task of paraphrasing a sentence to match another author's style. The dataset we use is a collection of Shakespeare's plays translated line by line into modern English. It was collected by Xu et al. (2012)[7] and used in prior work on supervised style transfer (Jhamtani et al., 2017). This is a parallel corpus and thus we follow the setting in the formality transfer task. We use $\mathcal{D}_1$ to denote modern English, and $\mathcal{D}_2$ to denote Shakespeare-style English. Results are shown in Table 1.

**Related Language Translation.** Next, we test our method on a challenging related language translation task (Pourdamghani & Knight, 2017; Yang et al., 2018). This task is a natural test bed for unsupervised sequence transduction since the goal is to preserve the meaning of the source sentence while rewriting it into the target language. For our experiments, we choose Bosnian (bs) and Serbian (sr) as the related language pairs. We follow Yang et al. (2018) to report BLEU-1 score on this task since BLEU-4 score is close to zero. Results are shown in Table 2.

**Unsupervised MT.** In order to draw connections with a related work on general unsupervised machine translation, we also evaluate on the WMT'16 German English translation task. This task is substantially more difficult than the style transfer tasks considered so far. We compare with the state-of-the-art UNMT system using the existing implementation from the XLM codebase,[8] and implement our approach in the same framework with XLM initialization for fair comparison. We train both systems on 5M non-parallel sentences from each language. Results are shown in Table 2.

In Tables 1 we also list the PPL of the test set under the external LM for both the source and target domain. PPL of system outputs should be compared to PPL of the test set itself because extremely low PPL often indicates that the generated sentences are short or trivial.

## 5.2 RESULTS

Tables 1 and 2 demonstrate some general trends. First, UNMT is able to outperform

Table 2: BLEU for decipherment, related language translation (Sr-Bs), and general unsupervised translation (En-De).

| Model | Decipher | Sr-Bs | Bs-Sr | En-De | De-En |
|-------|----------|-------|-------|-------|-------|
| Shen et al. (2017) | 50.8 | - | - | - | - |
| Yang et al. (2018) | 49.3 | 31.0 | 33.4 | - | - |
| UNMT | 76.4 | 31.4 | 33.4 | 26.5 | 32.2 |
| BT+NLL | 78.0 | 29.6 | 31.4 | - | - |
| Ours | **78.4** | **36.2** | **38.3** | 26.9 | 32.0 |

other prior methods in unsupervised text style transfer, such as (Yang et al., 2018; Hu et al., 2017; Shen et al., 2017). The performance improvements of UNMT indicate that flexible and powerful

---

[7]https://github.com/tokestermw/tensorflow-shakespeare
[8]https://github.com/facebookresearch/XLM

architectures are crucial (prior methods generally do not have an attention mechanism). Second, our model achieves comparable classification accuracy to UNMT but outperforms it in all style transfer tasks in terms of the reference-BLEU, which is the most important metric since it directly measures the quality of the final generations against gold parallel data. This indicates that our method is both effective and consistent across many different tasks. Finally, the BT+NLL baseline is sometimes quite competitive, which indicates that the addition of a language model alone can be beneficial. However, our method consistently outperforms the simple BT+NLL method, which indicates the effectiveness of the additional entropy regularizer in Eq. 5 that is the byproduct of our probabilistic formulation.

Next, we examine the PPL of the system outputs under pretrained domain LMs, which should be evaluated in comparison with the PPL of the test set itself. For both the sentiment transfer and the formality transfer tasks in Table 1, BT+NLL achieves extremely low PPL, lower than the PPL of the test corpus in the target domain. After a close examination of the output, we find that it contains many repeated and overly simple outputs. For example, the system generates many examples of "I love this place" when transferring negative to positive sentiment (see Appendix A.3 for examples). It is not surprising that such a trivial output has low perplexity, high accuracy, and low BLEU score. On the other hand, our system obtains reasonably competitive PPL, and our approach achieves the highest accuracy and higher BLEU score than the UNMT baseline.

## 5.3 FURTHER ABLATIONS AND ANALYSIS

**Parameter Sharing.** We also conducted an experiment on the word substitution decipherment task, where we remove parameter sharing (as explained in Section 3.2) between two directions of transduction distributions, and optimize two encoder-decoder instead. We found that the model only obtained an extremely low BLEU score and failed to generate any meaningful outputs.

**Performance vs. Domain Divergence.** Figure 3 plots the relative improvement of our method over UNMT with respect to accuracy of a naive Bayes' classifier trained to predict the domain of test sentences. Tasks with high classification accuracy likely have more divergent domains. We can see that for decipherment and en-de translation, where the domains have

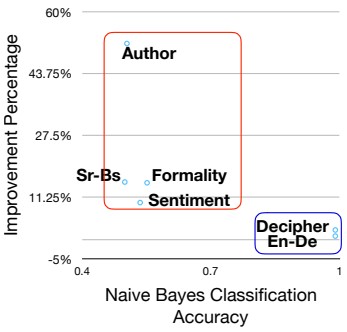

Figure 3: Improvement over UNMT vs. classification accuracy.

different vocabularies and thus are easily distinguished, our method yields a smaller gain over UNMT. This likely indicates that the (discrimination) regularization effect of the LM priors is less important or necessary when the two domains are very different.

**Why does the proposed model outperform UNMT?** Finally, we examine in detail the output of our model and UNMT for the author imitation task. We pick this task because the reference outputs for the test set are provided, aiding analysis. Examples shown in Table 3 demonstrate that UNMT tends to make overly large changes to the source so that the original meaning is lost, while our method is better at preserving the content of the source sentence. Next, we quantitatively examine the outputs from UNMT and our method by comparing the F1 measure of words bucketed by their syntactic tags. We use the open-sourced compare-mt tool (Neubig et al., 2019), and the results are shown in Figure 4. Our system has outperforms UNMT in all word categories. In particular, our system is much better at generating nouns, which likely leads to better content preservation.

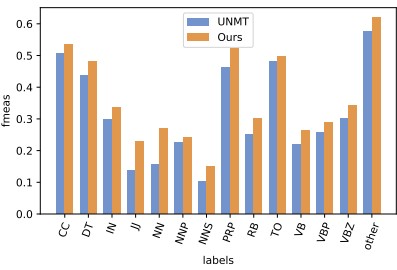

Figure 4: Word F1 score by POS tag.

Table 3: Examples for author imitation task

| Methods | Shakespeare to Modern |
|---|---|
| Source | Not to his father's . |
| Reference | Not to his father's house . |
| UNMT | Not to his brother . |
| Ours | Not to his father's house . |
| Source | Send thy man away . |
| Reference | Send your man away . |
| UNMT | Send an excellent word . |
| Ours | Send your man away . |
| Source | Why should you fall into so deep an O ? |
| Reference | Why should you fall into so deep a moan ? |
| UNMT | Why should you carry so nicely , but have your legs ? |
| Ours | Why should you fall into so deep a sin ? |

Table 4: Comparison of gradient approximation on the sentiment transfer task.

| Method | train ELBO↑ | test ELBO↑ | Acc. | $BLEU_r$ | $BLEU_s$ | $PPL_{\mathcal{D}_1}$ | $PPL_{\mathcal{D}_2}$ |
|---|---|---|---|---|---|---|---|
| Sample-based | -3.51 | -3.79 | 87.90 | 13.34 | 33.19 | 24.55 | 25.67 |
| Greedy | -2.05 | -2.07 | 87.90 | 18.67 | 48.38 | 27.75 | 35.61 |

Table 5: Comparison of gradient propagation method on the sentiment transfer task.

| Method | train ELBO↑ | test ELBO↑ | Acc. | $BLEU_r$ | $BLEU_s$ | $PPL_{\mathcal{D}_1}$ | $PPL_{\mathcal{D}_2}$ |
|---|---|---|---|---|---|---|---|
| Gumbel Softmax | -2.96 | -2.98 | 81.30 | 16.17 | 40.47 | 22.70 | 23.88 |
| REINFORCE | -6.07 | -6.48 | 95.10 | 4.08 | 9.74 | 6.31 | 4.08 |
| Stop Gradient | -2.05 | -2.07 | 87.90 | 18.67 | 48.38 | 27.75 | 35.61 |

**Greedy vs. Sample-based Gradient Approximation.** In our experiments, we use greedy decoding from the inference network to approximate the expectation required by ELBO, which is a biased estimator. The main purpose of this approach is to reduce the variance of the gradient estimator during training, especially in the early stages when the variance of sample-based approaches is quite high. As an ablation experiment on the sentiment transfer task we compare greedy and sample-based gradient approximations in terms of both train and test ELBO, as well as task performance corresponding to best test ELBO. After the model is fully trained, we find that the sample-based approximation has low variance. With a single sample, the standard deviation of the EBLO is less than 0.3 across 10 different test repetitions. All final reported ELBO values are all computed with this approach, regardless of whether the greedy approximation was used during training. The reported ELBO values are the evidence lower bound per word. Results are shown in Table 4, where the sampling-based training underperforms on both ELBO and task evaluations.

## 5.4 COMPARISON OF GRADIENT PROPAGATION METHODS

As noted above, to stabilize the training process, we stop gradients from propagating to the inference network from the reconstruction loss. Does this approach indeed better optimize the actual probabilistic objective (i.e. ELBO) or only indirectly lead to improved task evaluations? In this section we use sentiment transfer as an example task to compare different methods for propagating gradients and evaluate both ELBO and task evaluations.

Specifically, we compare three different methods:

- **Stop Gradient:** The gradients from reconstruction loss are not propagated to the inference network. This is the method we use in all previous experiments.

- **Gumbel Softmax (Jang et al., 2017):** Gradients from the reconstruction loss are propagated to the inference network with the straight-through Gumbel estimator.

- **REINFORCE (Sutton et al., 2000):** Gradients from reconstruction loss are propagated to the inference network with ELBO as a reward function. This method has been used in previous work for semi-supervised sequence generation (Miao & Blunsom, 2016; Yin et al., 2018), but often suffers from instability issues.

We report the train and test ELBO along with task evaluations in Table 5, and plot the learning curves on validation set in Figure 5.[9] While being much simpler, we show that the stop-gradient trick produces superior *ELBO* over Gumbel Softmax and REINFORCE. This result suggests that stopping gradient helps better optimize the likelihood objective under our probabilistic formulation in comparison with other optimization techniques that propagate gradients, which is counter-intuitive. A likely explanation is that as a gradient estimator, while clearly biased, stop-gradient has substantially reduced variance. In comparison with other techniques that offer reduced bias but extremely high variance when applied to our model class (which involves discrete sequences as latent variables), stop-gradient actually leads to better optimization of our objective because it achieves better balance of bias and variance overall.

---

[9]We remove REINFORCE from this figure since it is very difficult to stabilize training and obtain reasonable results (e.g. the ELBO value is much worse than others in Table 5)

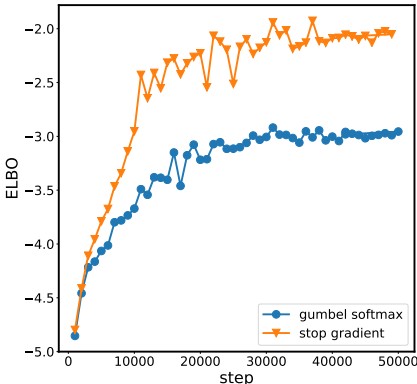

Figure 5: ELBO on the validation set v.s. the number training steps.

## 6  CONCLUSION

We propose a probabilistic generative forumalation that unites past work on unsupervised text style transfer. We show that this probabilistic formulation provides a different way to reason about unsupervised objectives in this domain. Our model leads to substantial improvements on five text style transfer tasks, yielding bigger gains when the styles considered are more difficult to distinguish.

## ACKNOWLEDGEMENT

The work of Junxian He and Xinyi Wang is supported by the DARPA GAILA project (award HR00111990063) and the Tang Family Foundation respectively. The authors would like to thank Zichao Yang for helpful feedback about the project.

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

# A    APPENDIX

## A.1    MODEL CONFIGURATIONS.

We adopt the following attentional encoder-decoder architecture for UNMT, BT+NLL, and our method across all the experiments:

- We use word embeddings of size 128.
- We use 1 layer LSTM with hidden size of 512 as both the encoder and decoder.
- We apply dropout to the readout states before softmax with a rate of 0.3.
- Following Lample et al. (2019), we add a max pooling operation over the encoder hidden states before feeding it to the decoder. Intuitively the pooling window size would control how much information is preserved during transduction. A window size of 1 is equivalent to standard attention mechanism, and a large window size corresponds to no attention. See Appendix A.2 for how to select the window size.
- There is a noise function for UNMT baseline in its denoising autoencoder loss (Lample et al., 2017; 2019), which is critical for its success. We use the default noise function and noise hyperparameters in Lample et al. (2017) when running the UNMT model. For BT+NLL and our method we found that adding the extra noise into the self-reconstruction loss (Eq. 4) is only helpful when the two domains are relatively divergent (decipherment and related language translation tasks) where the language models play a less important role. Therefore, we add the default noise from UNMT to Eq. 4 for decipherment and related language translation tasks only, and do not use any noise for sentiment, author imitation, and formality tasks.

## A.2    HYPERPARAMETER TUNING.

We vary pooling windows size as $\{1, 5\}$, the decaying patience hyperparameter $k$ for self-reconstruction loss (Eq. 4) as $\{1, 2, 3\}$. For the baseliens UNMT and BT+NLL, we also try the option of not annealing the self-reconstruction loss at all as in the unsupervised machine translation task (Lample et al., 2018). We vary the weight $\lambda$ for the NLL term (BT+NLL) or the KL term (our method) as $\{0.001, 0.01, 0.03, 0.05, 0.1\}$.

## A.3    SENTIMENT TRANSFER EXAMPLE OUTPUTS

We list some examples of the sentiment transfer task in Table 6. Notably, the BT+NLL method tends to produce extremely short and simple sentences.

## A.4    REPETITIVE EXAMPLES OF BT+NLL

In Section 5 we mentioned that the baseline BT+NLL has a low perplexity for some tasks because it tends to generate overly simple and repetitive sentences. From Table 1 we see that two representative tasks are sentiment transfer and formatliy transfer. In Appendix A.3 we have demonstrated some examples for sentiment transfer, next we show some repetitive samples of BT+NLL in Table 7.

Table 6: Random Sentiment Transfer Examples

| Methods | negative to positive |
|---|---|
| Original | the cake portion was extremely light and a bit dry . |
| UNMT | the cake portion was extremely light and a bit spicy . |
| BT+NLL | the cake portion was extremely light and a bit dry . |
| Ours | the cake portion was extremely light and a bit fresh . |
| | |
| Original | the " chicken " strip were paper thin oddly flavored strips . |
| UNMT | the " chicken " were extra crispy noodles were fresh and incredible . |
| BT+NLL | the service was great . |
| Ours | the " chicken " strip were paper sweet & juicy flavored . |
| | |
| Original | if i could give them a zero star review i would ! |
| UNMT | if i could give them a zero star review i would ! |
| BT+NLL | i love this place . |
| Ours | i love the restaurant and give a great review i would ! |
| | positive to negative |
| Original | great food , staff is unbelievably nice . |
| UNMT | no , food is n't particularly friendly . |
| BT+NLL | i will not be back . |
| Ours | no apologies , staff is unbelievably poor . |
| | |
| Original | my wife and i love coming here ! |
| UNMT | my wife and i do n't come here ! |
| BT+NLL | i will not be back . |
| Ours | my wife and i walked out the last time . |
| | |
| Original | my wife and i love coming here ! |
| UNMT | my wife and i do n't come here ! |
| BT+NLL | i will not be back . |
| Ours | my wife and i walked out the last time . |
| | |
| Original | the premier hookah lounge of las vegas ! |
| UNMT | the worst museum of las vegas ! |
| BT+NLL | the worst frame shop of las vegas ! |
| Ours | the hallways scam lounge of las vegas ! |

Table 7: Repetitive examples of BT+NLL baseline on Formality transfer.

| Original | Transferred |
|---|---|
| formal to informal | |
| I like Rhythm and Blue music . | I like her and I don't know . |
| There's nothing he needs to change . | I don't know , but I don't know . |
| I enjoy watching my companion attempt to role @-@ play with them . | I don't know , but I don't know . |
| I am watching it right now . | I don't know , but I don't know . |
| That is the key point , that you fell asleep . | I don't know , but I don't know . |
| informal to formal | |
| its a great source just download it . | I do not know , but I do not know . |
| Happy Days , it was the coolest ! | I do not know , but I do not know . |
| I used to play flute but once I started sax , I got hooked . | I do not know , but I do not know . |
| The word you are looking for is ............. strengths | The word you are looking for is : ) |
| Plus you can tell she really cared about her crew . | Plus you can tell she really cared about her crew . |

