# OpenReview forum: "A Probabilistic Formulation of Unsupervised Text Style Transfer"
_ICLR.cc/2020/Conference — Accept (Spotlight)_

### Official Review · AnonReviewer3 · 2019-10-20
**Official Blind Review #3**

**Rating:** 6

**Review:**

The main contribution of this paper is a principled probabilistic framework of unsupervised sequence to sequence transfer (text to text in particular).

However, I believe there is a large disconnect between the probabilistic formulation written it section 3 and whats actually happening experimentally in section 5. It is not clear whether the model is *actually* optimizing an ELBO because the gradients from sequence reconstruction loss are not backpropogated to the inference network as explained in paragraph on Approximating Gradients of ELBO. Moreover this restriction makes the authors method almost the same as the one used for unsupervised neural machine translation by Lample et al 2017 and Artetxe et al 2017. I would like to see a more detailed analysis from authors on how far the performance of Gumbel-softmax estimator and REINFORCE estimator is from simple stop-gradient estimator used in experiments.

In terms of experimental setup I like that the authors considered a large suite of experiments across various tasks. Although the evaluation metrics on text style transfer tasks like sentiment transfer, formality transfer, author imitation are in line with previous work ideally the human evaluation needs to be done to truly see how well each method performs. On unsupervised machine translation, authors show a large improvement on Serbian-Bostian translation. I am a bit skeptical since as I wrote above the proposed method is very similar to previously proposed unsupervised neural machine translation approaches and it is not clear why we are seeking such a large gain of 5 BLEU points.

Overall I think it is a well written paper with a large experimental suite, although I am skeptical of actual connection between probabilistic formulation and whats actually happening in practice.

================================================
Update: I have raised the score from 3 to 6.

**Experience Assessment:**

I have published in this field for several years.

**Review Assessment: Checking Correctness Of Derivations And Theory:**

I carefully checked the derivations and theory.

**Review Assessment: Checking Correctness Of Experiments:**

I assessed the sensibility of the experiments.

**Review Assessment: Thoroughness In Paper Reading:**

I read the paper at least twice and used my best judgement in assessing the paper.

---

> ### Author Response · Authors · 2019-11-15
> **Response to Reviewer #3**
>
> Thank you for the time and comments!
>
> ##  Q1: The disconnect between the proposed probabilistic model and what’s actually happening ?
> Thank you for bringing this up! This is an extremely important point and is something we believe can be cleared up -- specifically, we believe we have supported the case in additional experiments (see below) that this is an optimization issue instead of modeling issue.
>
> Stop-gradient can be viewed as an approximation technique for optimizing the training objective of the proposed probabilistic model. As you mention, many techniques fall into this category: Gumbel softmax approximates the true gradient with a biased estimator, hoping to reduce variance. Other techniques yield unbiased estimators, but at the cost of higher variance. Finding an effective optimization technique for a given model class often involves some degree of empirical exploration of the bias-variance tradeoff. Stop-gradient, when viewed in this light, is certainly a biased estimator of the true gradient, but may substantially reduce variance. We completely agree that this point could be supported more effectively with further experiments and comparisons. Specifically, is stop-gradient actually a better optimizer for our model class than Gumbel softmax or REINFORCE? Or does stop-gradient just lead to better task performance without actually better optimizing the modeling objective we claim to care about? If the latter were true, your concern about a disconnect between model and training procedure would be well-founded.
>
> In order to resolve this, we have run additional experiments on the sentiment transfer task. We compare stop-gradient, Gumbel softmax, and REINFORCE as optimization techniques, reporting the best train and test ELBO under our model class achieved with each approach -- as well as task performance corresponding to best test ELBO. These results are presented in Table 5 and discussed in Section 5.4 of the updated paper draft.
>
> The key finding is that stop-gradient leads to better training and test ELBO in our model class than propagating gradients with either REINFORCE or Gumbel softmax, validating stop-gradient as a better choice if our goal is truly to optimize our models training objective. (In other words, whichever optimization method achieves the better ELBO can be seen as a superior method for optimizing the proposed fully probabilistic model.) Further, across optimization methods, ELBO is correlated with task performance -- i.e. for these three optimization methods, the better the final ELBO achieved, the better the task performance.
>
> Together, we hope these results help support the case that, while somewhat unsatisfying a priori as a gradient estimator, the use of stop-gradient is in fact about optimization in our actual model class. If you think it would help support the case further, we can add similar experiments on the other tasks in future revisions. We also believe that one nice thing about our probabilistic formulation is that it allows us to separate out problems of learning the model and optimization, as we did here. This could help guide future work in better ways to optimize such probabilistic objectives.
>
>
>
> ## Q2: Similarities to unsupervised neural machine translation
> Yes, we agree that the underlying ELBO objective for our model class is similar to non-ELBO training objectives used in related unsupervised MT systems. However, we believe this is partially a strength: One goal of this paper is a probabilistic formulation that relates and interprets prior work. That said, there are important distinctions between our ELBO objective and objectives used in related work. For example, the main difference between the proposed model and UNMT is the added language model and the KL loss term. As mentioned in the paper, the language model is more useful when the two domains are less divergent, where the language models behave like a discriminator to avoid copying, and copying is very likely to happen in this case without supervised data. Therefore, the proposed model performs similarly to UNMT on decipherment where the vocabulary from two domains are completely different. For close language translation a portion of the vocabulary is shared between domains, and the language model plays a bigger role yielding improved performance.

---

### Official Review · AnonReviewer2 · 2019-10-22
**Official Blind Review #2**

**Rating:** 6

**Review:**

In this paper, the authors propose a probabilistic framework for unsupervised text style transfer. Given two non-parallel corpora X,Y in different domains, the authors introduce unobserved corpora \bar{X}, \bar{Y}. These are used as latent variables that control the generation of the observed data. To train models, the paper proposes to optimize the evidence lower bound of the log marginal likelihood. To facilitate training, multiple techniques are suggested, such as parameter sharing, some gradient approximations and initialization with a reconstruction objective. The approach is evaluated on five style transfer tasks, as well as unsupervised machine translation. Models are evaluated with multiple metrics, and generally obtain reasonably strong performance.

I lean towards the acceptance of the paper because the approach is fairly simple and elegant, while obtaining promising results. The connections to back-translation and language models are also potentially interesting. However, while the paper aims to suggest a principled approach to style transfer, using greedy samples biases the reconstruction objective, and as such the method does not really optimize the ELBO.

Casting style transfer as data completion is a straight-forward idea that doesn't introduce unnecessary or too simplistic assumptions. Optimizing the ELBO follows naturally, and can lead to more diverse outputs than the BT+NLL approach, which misses the negative entropy term. Reference BLEU scores on all tasks are competitive, and sometimes clearly better, with strong baselines.

Greedily sampling latent sequences during training should ideally be justified more carefully as it biases the objective function. In particular, an experimental comparison to stochastic sampling, which should more closely approximate the expectation, would be appreciated. Additionally, detailing the similarities and differences between the proposed approach and current UNMT techniques could be helpful to some readers.

Questions:

Could you present the validation and test evidence lower bounds? If so, how is sampling performed?

In footnote 2, you mention tuning the strength of the KL regularizer. As the KL can be decomposed into 2 terms (Eq. 5), would it be beneficial to control each term separately?

**Experience Assessment:**

I have read many papers in this area.

**Review Assessment: Checking Correctness Of Derivations And Theory:**

I assessed the sensibility of the derivations and theory.

**Review Assessment: Checking Correctness Of Experiments:**

I assessed the sensibility of the experiments.

**Review Assessment: Thoroughness In Paper Reading:**

I read the paper at least twice and used my best judgement in assessing the paper.

---

> ### Author Response · Authors · 2019-11-15
> **Response to Reviewer #2**
>
> We thank the reviewer for the time and comments. Due to time limitations we could only address major points, but we’ll make sure to reflect all advice in future revisions.
>
> ## Q1: Comparison of greedy and sampling decoding in terms of ELBO
> We agree that a comparison between gradient approximation techniques in this context would be informative. In preliminary experiments not included in the paper draft we tried various approaches -- in retrospect (and based on your comment) we realize it makes complete sense to include this analysis. Thanks for the suggestion!
>
> We have updated the paper with a comparison between greedy and sample-based gradient approximations for ELBO on the sentiment transfer task. Please see the last paragraph in Section 5.3 and Table 4 for details. Here we report training ELBO, test ELBO, and task performance. We find that the greedy approximation leads to better optimization of both training and test ELBO and better task performance. It’s worth noting that once the model is trained, the sample-based approximation of ELBO is low-varaince. Thus, in Table 4, we are showing the sample-based training ELBO regardless of the gradient approximation technique. The fact that the greedy gradient approximation leads better ELBO optimization even though the greedy estimator is biased indicates that the sampled-based approximation (which is unbiased) has much higher variance during the early stages of learning -- we are trading variance for bias with positive effect. Using more samples might mitigate this issue during training, but would require substantially more computation. We are currently running additional experiments to explore methods for reducing training variance with sample-based approximations, and hope to include these results in future revisions.
>
> ## Q2: Would it be beneficial to control each term in KL separately?
> This is a good point, and we believe that it would be beneficial with more control for each term. However, this also introduces an additional tunable hyperparameter. Due to time limitation we are not able to fully verify this hypothesis on these tasks, but we will try to include experimental analysis regarding this separate control in the next revision.

---

### Official Review · AnonReviewer1 · 2019-10-25
**Official Blind Review #1**

**Rating:** 8

**Review:**

Summary:

This paper introduces a probabilistic generative model for
unsupervised style transfer of text. The approach introduced in
the paper does not require paired training data. An
encoder-decoder model is trained to transfer text from one style
to another and back.

Review:

This work is very well-written and easy to follow. The
contribution is clearly articulated as while there are
probabilistic generative models for transfer in the
literature (Shen et al does include one) they don't perform as
well. Ablation studies further confirm the need for the
particular kind of parameter sharing used in the model in the
paper. Great results are shown on 5 text transfer problems.

Clarifications and improvements:

Just for clarity, in the last paragraph on page 4. It says two encoder-decoder
models are learnt, but isn't the idea that there is effectively only one
encoder and one decoder learned that just put together in different ways
during training? I'm also curious why the baseline of BT+NLL was so strong? Is
having the loss of a language model work that much better than the regular
entropy term?

I would also like if possible if you could share some of the repetitive examples
created by BT-NLL which explain its low PPL.

**Experience Assessment:**

I have read many papers in this area.

**Review Assessment: Checking Correctness Of Derivations And Theory:**

N/A

**Review Assessment: Checking Correctness Of Experiments:**

I assessed the sensibility of the experiments.

**Review Assessment: Thoroughness In Paper Reading:**

N/A

---

> ### Author Response · Authors · 2019-11-15
> **Response to Reviewer #1**
>
> We thank the reviewer for the useful feedback and clarification questions!
>
> ## Q1: Isn't the idea that there is effectively only one encoder and one decoder learned that just put together in different ways during training?
> Yes, you are correct! Thanks for pointing out the possible confusion. We will clarify this point in the paper. Due to parameter sharing, during training we learn one shared encoder and one shared decoder.
>
> ## Q2: Why is the BT + NLL baseline so strong ?
> Apart from the decipherment task, BT + NLL actually underperforms UNMT, which does not have the language model. This is most pronounced in the sentiment and formality transfer tasks where BT + NLL fails with very low perplexity. Therefore, in all cases but decipherment, it seems that adding a language model without the complete KL term (equation 5) does not result in superior performance.
>
> Thanks for the suggestion to include examples of repetitive outputs! There are several examples from BT + NLL on sentiment transfer in A.3. We have we added additional repetitive examples on the formality transfer task in Appendix A.4 (Table 7).

---

### Author Response · Authors · 2019-11-15
**Revision Submitted**

We have submitted a revised manuscript and made the following modifications to address the reviewers' major concerns:

-- Compared sampling decoding and greedy decoding as different approximation methods, in terms of both ELBO and task performance (the last paragraph of Section 5.3)
-- Compared different gradient propagation methods, in terms of both ELBO and task performance (Section 5.4)


While limited by time in the response period, we do still plan to address *all* the reviewer’s comments in future revisions. We also welcome any further feedbacks to improve this paper !

---

### Decision · Program_Chairs · 2019-12-19

**Decision:**

Accept (Spotlight)

**Comment:**

This paper proposes an unsupervised text style transfer model which combines a language model prior with an encoder-decoder transducer. They use a deep generative model which hypothesises a latent sequence which generates the observed sequences. It is trained on non-parallel data and they report good results on unsupervised sentiment transfer, formality transfer, word decipherment, author imitation, and machine translation. The authors responded in depth to reviewer comments, and the reviewers took this into consideration. This is a well written paper, with an elegant model and I would like to see it accepted at ICLR.